# *Salmonella* Phage CKT1 Effectively Controls the Vertical Transmission of *Salmonella* Pullorum in Adult Broiler Breeders

**DOI:** 10.3390/biology12020312

**Published:** 2023-02-14

**Authors:** Ketong Cui, Peiyong Li, Jiaqi Huang, Fang Lin, Ruibo Li, Dingguo Cao, Guijuan Hao, Shuhong Sun

**Affiliations:** 1Department of Preventive Veterinary Medicine, College of Animal Science and Technology, Shandong Agricultural University, Tai’an 271018, China; 2Shandong Provincial Key Laboratory of Animal Biotechnology and Disease Control and Prevention, Shandong Agricultural University, Tai’an 271018, China; 3Poultry Research Institute of Shandong Academy of Agricultural Sciences, Jinan 250000, China

**Keywords:** *Salmonella* Pullorum, phage therapy, vertical transmission, poultry, reproductive infection

## Abstract

**Simple Summary:**

*Salmonella enterica* serovar Gallinarum biovar Pullorum (*S.* Pullorum), a septicemic pathogenic bacterium of poultry due to high mortality of chicks or poults, could not only result in a high mortality rate among embryos and chicks, but also lead to persistent infection and transmission to eggs or progeny in adult chickens during the carrier state. Diseases caused by *Salmonella* infection via vertical transmission continue to pose a highly significant threat to both human and animals; however, research on phage therapy regarding the vertical transmission of *Salmonella* is currently lacking. In this study, we evaluated the effect of *Salmonella* phage CKT1 on controlling the vertical infection of *S.* Pullorum by detecting the bacterial load in the reproductive system, eggshells and liquid whole eggs of adult broiler breeders. The results showed that the oral use of phage CKT1 could effectively reduce the colonization level of *S.* Pullorum in various organs including liver, spleen, heart, ovary and oviduct of adult breeders. More importantly, phage treatment significantly decreased the *Salmonella*-specific IgG level in serum of infected chickens, and reduced the bacterial load in eggshells and liquid whole eggs, thus decreasing the probability of vertical transmission of *S.* Pullorum in poultry. In addition, the total bacterial load in the poultry breeding environment was also significantly reduced after phage therapy, probably due to the decrease in the *Salmonella* load in cecal contents and restoration of the intestinal microbiome in chickens.

**Abstract:**

Phage therapy is widely being reconsidered as an alternative to antibiotics for the treatment of multidrug-resistant bacterial infections, including salmonellosis caused by *Salmonella*. As facultative intracellular parasites, *Salmonella* could spread by vertical transmission and pose a great threat to both human and animal health; however, whether phage treatment might provide an optional strategy for controlling bacterial vertical infection remains unknown. Herein, we explored the effect of phage therapy on controlling the vertical transmission of *Salmonella enterica* serovar Gallinarum biovar Pullorum (*S.* Pullorum), a poultry pathogen that causes economic losses worldwide due to high mortality and morbidity. A *Salmonella* phage CKT1 with lysis ability against several *S. enterica* serovars was isolated and showed that it could inhibit the proliferation of *S.* Pullorum in vitro efficiently. We then evaluated the effect of phage CKT1 on controlling the vertical transmission of *S.* Pullorum in an adult broiler breeder model. The results demonstrated that phage CKT1 significantly alleviated hepatic injury and decreased bacterial load in the liver, spleen, heart, ovary, and oviduct of hens, implying that phage CKT1 played an active role in the elimination of *Salmonella* colonization in adult chickens. Additionally, phage CKT1 enabled a reduction in the *Salmonella*-specific IgG level in the serum of infected chickens. More importantly, the decrease in the *S.* Pullorum load on eggshells and in liquid whole eggs revealed that phage CKT1 effectively controlled the vertical transmission of *S.* Pullorum from hens to laid eggs, indicating the potential ability of phages to control bacterial vertical transmission.

## 1. Introduction

In the poultry industry, *Salmonella enterica* serovar Gallinarum biovar Pullorum (*S.* Pullorum) can produce an acute systemic disease with severe gastrointestinal illness in young chicks followed by a high mortality rate. However, it rarely causes such severe clinical disease in adult chickens [1]. A small number of *Salmonella enterica* serovars can cause systemic disease in a restricted range of host species and lead to carriers continuing to shed *Salmonella* for years in the absence of clinical diseases, such as *S. typhi* in humans, *S. dublin* in cattle and *S.* Pullorum in chickens [1,2,3]. *S.* Pullorum may persist for long periods in convalescent chickens, with colonization in the macrophages and reproductive tract of hens at sexual maturity, thus resulting in infected progeny by trans-ovarian and trans-oviductal transmission leading to the direct contamination of yolk, albumen, eggshell membranes, or eggshells [4]. Infected chickens also spread the pathogen persistently to other environments via the fecal–oral route, resulting in the contamination of hatching eggs caused by contaminated feces penetrating the eggshell after or during oviposition [5]. Over the years, *S.* Pullorum eradication programs, especially for broiler breeders, have been undertaken in many countries [6]. Although some developed countries have essentially eliminated *S.* Pullorum by serological testing, pullorum disease remains a major threat in some developing countries where the intensification of the poultry industry is in its infancy [7,8].

In recent years, many countries have banned the use of antibiotics as growth promoters in animal feed [9], which has led to a renewed interest in phage therapy [9]. Previous research has proved that phage therapy can block the horizontal transmission of *S. gallinarum* [10], thus bacterial re-isolation from the organs and mortality decreased significantly in both challenged and contact chickens treated with the bacteriophages compared with untreated chickens [9,11]. Vertical transmission may lead to high mortality in chicks and the contamination of eggs as well as chicken meat, which has become a worldwide public health concern. *S.* Pullorum infection in the genital tract may make phage therapy difficult due to the complex route of drug transportation [12], and lower bacterial colonization during chronic infection of adult chickens may also be a disadvantage of the interaction between phages and hosts [13]. However, *S.* Pullorum persisting in the intestine and both the liver and spleen likely provides a bridge for phages to reach the genital tract to lyse bacteria and produce progeny phages [14]. This provides a possibility for a phage to treat genital tract infection and control vertical transmission. Diseases caused by *S.* Pullorum infection via vertical transmission continue to pose a highly significant threat to poultry; however, research on phage therapy regarding the vertical transmission of *S.* Pullorum is currently lacking.

Therefore, this work aimed to explore the effect of phage therapy on controlling the vertical transmission of *S.* Pullorum by detecting the bacterial load in the ovary, oviduct, eggshells, and liquid whole eggs of adult broiler breeders, which may contribute to a feasible strategy for protecting young chicks from vertical infection.

## 2. Materials and Methods

### 2.1. Bacterial Strains and Growth Conditions

All bacterial strains in this study were listed in Appendix A Table A1. The standard *S.* Pullorum strain (CVCC526) was purchased from the China Veterinary Culture Collection Center (CVCC, Beijing, China), and the other strains were isolated from chicken farms in China and stored in the lab. All *Salmonella* strains stored at −70 °C were streaked onto Xylose Lysine Deoxycholate (XLD) agar plate (Haibo Biotechnology, Qingdao, China) and incubated at 37 °C for 24 h for subsequent testing, while other bacterial strains were streaked on Luria–Bertani (LB) agar plates.

### 2.2. Phage Isolation and Purification

*S.* Pullorum CVCC526 was used for phage isolation and purification. A modified method was used to isolate *Salmonella* phages using farm-sewage samples [15]. Briefly, sewage suspensions were centrifuged at 10,000 rpm for 10 min to remove particulates, then the supernatants were sterilized by 0.22 μm membrane filters. 10 mL of the above sewage samples were added into 10 mL of sterile double-strength LB broth containing 2 mM CaCl_2_ and approximately 10^7^ colony-forming unit (CFU)/mL of *Salmonella* cells, incubating at 37 °C, 80 rpm for 12 h. Crude lysates were evaluated for plaque formation by the double-agar overlay assay after centrifugation and chloroform treatment. Then, a single plaque was picked from the double-layer agar into SM buffer (10 mM MgSO_4_, 100 mM NaCl, 0.01% *w*/*v* gelatin, 50 mM Tris-HCl, pH 7.5), and then diluted and plated onto the seeded double-agar. The phages were purified using five consecutive rounds from single plaques and then resuspended with SM buffer and stored at 4 °C for use. For phage propagation, freshly grown cultures of *S.* Pullorum CVCC526 (approximately 10^8^ CFU/mL) inoculated with phages at an MOI of 0.001 were incubated at 37 °C for 4 h, then phage suspensions were filtered using a 0.22-μm-pore-size membrane. The number of phages was determined by serial dilution and plating on the seeded top agar plates.

### 2.3. Transmission Electron Microscopy

Freshly high-titer phages were prepared accordingly to the above method to obtain phage pellets with titers higher than 10^10^ plaque forming units (PFU)/mL. Then, 10 μL of samples were dropped onto a copper grid and negatively stained with 2% phosphor-tungstic acid, and then the grids were examined using an H7650 transmission electron microscope (TEM) (Hitachi, Tokyo, Japan).

### 2.4. Phage Characterization Assay

A total of 30 strains of bacteria were used to detect the phage host spectrum (Appendix A Table A1). All bacterial strains were tested for their susceptibility to the phage using spotting methods as previously described [16]. Phage adsorption ability was determined accordingly to a described previously method [17]. Briefly, phages in SM buffer (~1 × 10^7^ PFU/mL) were incubated with *S.* Pullorum CVCC526 (~1 × 10^8^ CFU/mL) at 37 °C in LB medium. Samples were withdrawn every 3 min and filtered with a 0.22 μm filter membrane immediately. The titer of unabsorbed phage particles in the filtrates was quantified by the double-agar overlay assay [15].

A modified method was used for the one-step growth curve assay [18]. Briefly, the phages (1 × 10^7^ PFU/mL) were incubated with *S.* Pullorum CVCC526 (~1 × 10^8^ CFU/mL) in 20 mL of LB medium at 37 °C for 9 min to allow maximum adsorption of phage; then the mixture was centrifuged at 4 °C to remove unabsorbed phage. The precipitated cells were resuspended with the same volume of fresh LB medium, and samples were incubated at 37 °C and taken every 10 min. The phage titer was quantified by double-agar overlay assay.

For the thermal sensitivity, the phage (~1 × 10^8^ PFU/mL) suspension in SM buffer was incubated at 37, 50, 60, 70, and 80 °C, respectively. Samples were taken at 30, 60 and 120 min, and then phage titer was quantified by the double-agar overlay assay. For the acid-base sensitivity, phages (~1 × 10^8^ PFU/mL) in SM buffer were incubated at 37 °C for 2 h in LB medium at pH ranging from 3 to 12, respectively.

For the lytic activity of phage in vitro, stationary bacterial cultures (~1 × 10^8^ CFU/mL) were mixed with the phage suspension in SM buffer at MOI of 0.0001, 0.001, 0.01, 0.1, 1 and 10, respectively, in LB medium, then cultures were incubated at 37 °C, 180 rpm for 8 h. Samples were taken every hour; then bacterial density was determined by the indication of an ultraviolet spectrophotometer at OD_600_. The phage titer was quantified by the double-agar overlay assay. All above experiments were performed with three independent repetitions.

### 2.5. Phage Genomic Analysis

Extraction and purification of phage genomic DNA were carried out using a lambda phage genomic DNA Kit (Zoman Biotek Corp., Beijing, China), according to the manufacturer’s protocol. The entire phage genome sequencing using Illumina NovaSeq PE150 was completed on the Illumina MiSeq second-generation sequencing platform at Beijing Novogene Bioinformatics Technology Co., Ltd. (Beijing, China). Open reading frames (ORFs) were predicted using GeneMarkS [19]. The National Center for Biotechnology Information (NCBI) Basic Local Alignment Search Tool (BLAST, http://blast.ncbi.nlm.nlm.nih.gov/, accessed on 21 January 2022) was applied for sequence similarity alignment, and the online annotation tool Rapid Annotation using Subsystem Technology (RAST, http://rast.nmpdr.org, accessed on 21 January 2022) was utilized for genome-wide alignment quick annotation [20]. BLAST was used to establish phylogenetic trees based on the sequence of the phage whole genome. Phage lifestyle was predicted using the PHACTS program [21]. The Virulence Factor Database (VFDB, http://www.mgc.ac.cn/VFs/main.htm, accessed on 23 January 2022) and Comprehensive Antibiotic Resistance Database (CARD, https://card.mcmaster.ca/, accessed on 23 January 2022) were queried to retrieve the toxic genes, virulent genes, and antibiotic-resistant genes in the phage genome, respectively [22,23].

### 2.6. Phage Treatment in Salmonella-Infected Adult Chickens

All animal work was reviewed and approved by the Laboratory Animal Care Committee of Shandong Agricultural University [permit number SDAUA-2021-034]. The T-cell response to *S.* Pullorum non-specific to mitogenic stimulation can fall sharply in hens at the onset of laying [24], which leads to re-proliferation of *S.* Pullorum at sexual maturity and persisting after sexual maturity, especially in the reproductive system [1,24]. Therefore, we try to establish an adult broiler chicken model with *S.* Pullorum CVCC526 by pectoral injection at the onset of sexual maturity.

A total of 20 *Salmonella*-free broiler breeders (Hubbard Efficiency Plus, female) at 22 weeks old were authorized by Yisheng (Yisheng Livestock and Poultry Breeding Co., Ltd., Shandong, China) and strictly transported to avoid any contamination. Chickens were randomly divided into two groups (10 chickens/group). Each group was housed in a separate 20-square-meter room more than 50 m apart with wood shavings and waterlines by different experimental personnel to avoid cross-contamination. All experimental chickens were determined to be free of *S.* Pullorum infection via weekly *Salmonella* isolation from anal swabs before this experiment [25]. Chickens were provided with sterile feed and water ad libitum. The Group 1 chickens were inoculated with *S.* Pullorum CVCC526 (1.8 × 10^9^ CFU) in 200 μL of PBS by injection into the breast muscle, and those in Group 2 were injected with PBS buffer only (Figure 1, time −3 week). Three weeks after *Salmonella* infection (day 0), the serum was collected from two groups of chickens, then the Group 1 chickens were divided into an *S.*P group (only *Salmonella* challenged) and an *S.*P + Phg group (*Salmonella* challenged and phage treated) with 5 chickens in each group, whereas Group 2 chickens without infection were divided into a Con group (control group) and a Phg group (only phage treated). To avoid any contamination of *Salmonella* or phages, the four groups were housed in separate air-filtered isolate cabinets (FENGSHI Group, Suzhou, China; instrument model: FS-GY size 2350 × 900 × 1750 mm) in a well-ventilated room with four compartments of approximately 10 square meters.

At the times designated as day 1 and day 2 (Figure 1), the chickens in the Phg group and *S.*P + Phg group were treated orally with a single dose of phage (1 × 10^8^ PFU/chicken) in a total volume of 1 mL, whereas the chickens in the Con group and *S.*P group were orally inoculated with the same volume of sterile SM buffer. All chickens were maintained at an age-appropriate temperature for 6 days and monitored daily for clinical signs and egg collection. At day 6 post phage treatment, all chickens from each group were weighed, and blood was collected from the brachial wing vein. Then, chickens were sacrificed by cervical dislocation, and the liver, spleen, heart, ovary, follicle, and oviduct were taken for subsequent enumeration of *Salmonella*.

### 2.7. Serum Assays

To determine the effect of phage treatment on the prevalence of *S.* Pullorum infection in adult breeders, the serums of hens at day 0 and day 6 (Figure 1) were collected for the detection of antibody level of *Salmonella*-specific IgG via an ELISA kit (ID Screen^®^ Avian *Salmonella* Indirect—Groups B and D, ID.VET, France) with the method described at http://www.ID.vet.com. Besides, the *S.* Pullorum-specific antibody was also determined by the slide agglutination test using commercialized agglutination antigens (Beijing Zhonghai Biotech Co. Ltd., Beijing, China).

### 2.8. Detection of Bacterial Load in Various Tissues, Eggshell and Liquid Whole Egg

To determine bacterial loads in the liver, spleen, heart, ovary, follicle, and oviduct, tissue samples were weighed and homogenized in 1 mL of PBS buffer [25]. The dilutions of the homogenates were plated onto XLD plates for the *S.* Pullorum load. Three suspected *S.* Pullorum colonies for each sample were identified by polymerase chain reaction (PCR) assays using a specific target gene *ipaJ* [26].

To quantify the *S.* Pullorum load in eggshells, the method developed by Al-Ajeeli et al. was used with a slight modification [27]. Briefly, the collected eggs were individually placed in sterile Whirl-Pak sample bags containing 20 mL PBS and gently massaged for 1 min. Then, 1 mL of liquid was transferred aseptically from each sample bag into a tube. Serial dilutions were plated on XLD plates to quantify bacterial load. In addition, 1 mL of the liquid whole egg was collected in tubes to determine the bacterial load via serial dilution and plating on XLD agar plates. After incubation at 37 °C for 24 h, *Salmonella* colonies were counted and identified per the method above.

### 2.9. Detection of Bacterial Load in Cecal Contents

The load of *S.* Pullorum in cecal contents was determined using real-time-quantitative polymerase chain reaction (qPCR) with *S.* Pullorum-specific *SEP1* gene primers (SEP1-Forward: 5′-CCCGGATTGGACCTCAAGTG-3′, SEP1-Reverse: 5′-ATGTTACGGGACGAGTG GGT-3′, SEP1-Probe: 5′-6-FAM-ACGCACAATCACTGTGCGACCATCCGG-BHQ1-3′). Total DNA was extracted from the cecal contents of all chickens at day 6 using Nabil et al.’s method with slight modifications [28]. Briefly, the cecal content suspension was incubated with proteinase K and lysis buffer at 56 °C for 10 min; then anhydrous ethanol was added to the lysate. Samples were washed and centrifuged, and DNA was eluted with elution buffer. qPCR was performed in a final volume of 20 µL containing 1 µL of DNA template, 10 µL of 2 × QuantiTect Probe RT-PCR Master Mix (Vazyme, Tianjin, China), 6.8 µL PCR grade water, 0.4 µL of each primer (50 pmol conc.), and 0.4 µL of each probe (30 pmol conc.). Primary denaturation was performed at 94 °C for 5 min, followed by 40 cycles of denaturation at 94 °C for 5 s, annealing at 60 °C for 15 s, and extension at 72 °C for 30 s. A known standard (of known CFU/g) was ten-fold serially diluted, and then DNA was extracted separately from 10^4^ to 10^8^ CFU of cells and tested by qPCR with the same conditions as the unknown samples. The standard was included in all PCR runs, and the *S.* Pullorum load in samples was quantified with CFU/g.

### 2.10. Detection of Bacterial Load in the Air

The total bacterial load in the air was quantified by the open-plate method with a slight modification [29]. Briefly, the total bacterial load in the air of the isolate cabinet on day 3 and day 6 after the use of phage was quantified by opening sterile 90 mm plates containing 20 mL of PBS buffer for 45 min. The plates were placed in three evenly distributed positions with three repetitions in the isolate cabinet. Then, 1 mL liquid was transferred aseptically from plates into tubes, diluted with PBS buffer, and plated on LB plates.

### 2.11. Statistical Analysis

GraphPad Prism 9.0.0 software was used for data analysis. The data of bacterial load in tissues, eggshell, and liquid whole egg were statistically analyzed by the Mann–Whitney test. For the bacterial load in the air and the ratio of organs to bodies, one-way analysis of variance (ANOVA) and post hoc Tukey’s test were performed. Significance was determined at *p* < 0.05.

## 3. Results

### 3.1. Characteristics of Phage CKT1

*S.* Pullorum lytic phage CKT1 was observed to produce clear plaques with diameters ranging from 4.0 to 5.0 mm in 8 h (Figure 2A). Host spectrum results demonstrated lytic activity against *Salmonella enterica* serovars Typhimurium, Enteritidis, and Pullorum (Table A1), but other species including *Escherichia coli* and *Klebsiella pneumoniae* were not able to be lysed. TEM analysis revealed that phage CKT1 had a polyhedral head with a diameter of approximately 60 nm and a non-contractile tail of approximately 120 nm (Figure 2B). We examined the adsorption ability of phage CKT1, which had an average adsorption rate of 76% at 3 min and 92% at 9 min (Figure 2C), indicating the high and rapid adsorption ability of phage CKT1. A short latent period and high burst size of phage can facilitate the effective killing of bacteria. The one-step growth curve of phage CKT1 showed a latent period of approximately 10 min and a burst period of 20 min, with an average burst size of 147 PFU/cell; then phage titer stayed around 10^10^ PFU/mL after 50 min infection (Figure 2D).

The thermal sensitivity tests showed that phage CKT1 was stable at 37~60 °C but decreased at 70 °C in 120 min. When the temperature increased to 80 °C, phage CKT1 was almost completely inactivated in 30 min (Figure 2E). The acid-base sensitivity tests showed that phage CKT1 maintained stability at pH 4–11 for 2 h, but its activity decreased rapidly at both pH 2–3 and pH 12–13 (Figure 2F).

*S.* Pullorum CVCC526 was infected with phage CKT1 at different MOIs to evaluate its antibacterial ability in vitro. Figure 2G illustrates that bacterial proliferation was efficiently inhibited by phage CKT1 in 8 h (*p* < 0.0001) even at MOI of 0.0001. However, at all tested MOIs, phage titer could be increased to approximately 10^10^ PFU/mL, indicating its strong replication ability and lytic activity against *S.* Pullorum and potential for phage therapy (Figure 2H).

### 3.2. Genomic Analysis of Phage CKT1

Genomic analysis showed that phage CKT1 genome is dsDNA with a total length of 40,923 bp (GenBank: OK143508.1), belonging to the *Siphoviridae* family. A total of 64 genes were identified; 16 have known functions, including encoding structural proteins, lyases, and replication (Figure 3A). Phage CKT1 is highly similar to *Salmonella* phage SHWT1 and closely related to several other previously isolated *Salmonella* phages (Figure 3B). In addition, putative integrase genes, toxin genes, virulence genes, and antibiotic resistance genes were not detected in the genome of phage CKT1, implying that it might be a virulent phage with therapeutic potential.

### 3.3. Effects of Phage Treatment on the Salmonella-Specific Antibody in Chickens

To examine the effect of phage treatment on the vertical transmission of *Salmonella*, we constructed an adult broiler chicken model with *S.* Pullorum CVCC526 by pectoral injection at the onset of sexual maturity (Figure 1). These chickens resumed laying eggs in −1 week after *Salmonella* challenge. Analysis of bacterial load in eggshells and liquid whole eggs showed that the positive rate of *S.* Pullorum was 67% (6/9) on eggshells and 44% (4/9) in liquid whole eggs (Table 1), supporting the vertical transmission of *S.* Pullorum from adult hens to eggs. In addition, we determined if the *Salmonella*-specific antibody via the agglutination test at day 0 (Figure 1); the results showed that all infected chickens were positive for *Salmonella* antibodies, indicating the persistent presence of *S.* Pullorum in vivo (Figure 4A).

To determine the effect of phage treatment on the prevalence of *S.* Pullorum infection in adult breeders, we compared the *Salmonella*-specific antibody IgG level in serum at day 0 (Figure 1, 0 d) before phage treatment and at day 6 post-therapy (Figure 1). As shown in Figure 4B, the *Salmonella*-specific IgG level of the *S.*P group remained stable; however, the IgG level in the *S.*P + Phg group was incredibly decreased, by 91.96% (*p* < 0.05). Likewise, the slide agglutination test showed that 100% of chickens were positive for *S.* Pullorum at 6 d, whereas the positive rate of *Salmonella*-infected chickens was surprisingly decreased to 40% at 6 d post-phage therapy (Figure 4A).

### 3.4. Effects of Phage Treatment on the Bacterial Load in Various Tissues

*S.* Pullorum colonization in the early stages of the infection usually occurs in the liver, spleen, heart, and cecum. To examine the effect of phage therapy on tissues, we first evaluated the ratio of the liver and spleen weight to the body weight. Compared to the control group, the spleen ratio in both the *S.*P group and the *S.*P + Phg group decreased without statistical significance (Figure 5A), indicating that infection with *S.* Pullorum caused minimal damage to the spleen of adult chickens. However, compared to that of the control group, the liver ratio of the *S.*P group decreased significantly (*p* < 0.01) (Figure 5B), while the liver ratio of the *S.*P + Phg group showed no significant difference. This indicated that the liver might be affected by colonized *S.* Pullorum and that phage CKT1 probably alleviated injury to the liver caused by *S.* Pullorum. In addition, the liver and spleen ratio of the Phg group increased compared to that of the control group (*p* > 0.05); this may indicate that phage CKT1 affects the spleen and liver of adult chickens.

Next, the effect of phage treatment on *Salmonella* colonization in the chickens’ liver, spleen, heart, and cecum during the early stages of infection was quantified. As shown in Figure 5C, compared to the *S.*P group, the *S.* Pullorum load in the liver, spleen and heart in the *S.*P + Phg group decreased by nearly 83.1%, 94.4% and 89.7%, respectively. The *S.* Pullorum load in the above three organs in the Con group and Phg group could not be detected. Thus, these data indicated that phage CKT1 could reduce the *S.* Pullorum load in tissues and might further alleviate injury to the liver and spleen. Then, we utilized qPCR to quantify the *S.* Pullorum load in cecal contents. As shown in Figure 5C, the cecal *S.* Pullorum load in the *S.*P + Phg group decreased significantly (*p* < 0.05) compared to the *S.*P group, indicating that phage CKT1 could also reduce colonization of *S.* Pullorum in the cecum of adult chickens.

### 3.5. Effects of Phage CKT1 on Bacterial Load in Reproductive System, Eggs, and Breeding Environment

*S.* Pullorum persists within the ovary and oviduct of hens after sexual maturity, which leads to the occurrence of vertical transmission. Thus, the effect of phage CKT1 to limit the vertical transmission of *S.* Pullorum was examined. As shown in Figure 6A, compared with that of the *S.*P group, phage treatment significantly reduced the *S.* Pullorum load in the ovary and oviduct from below the detection line, and no *S.* Pullorum could be re-isolated from the ovary and oviduct in the *S.*P + Phg group, suggesting the elimination of *S.* Pullorum in the reproductive system with the use of phage CKT1. No *S.* Pullorum contamination was detected in both the Con and Phg groups.

For *S.* Pullorum in eggs, there is likely to be a failed hatching process and vertically infected chicks; therefore, we also detected the *S.* Pullorum load in liquid whole eggs. As shown in Figure 6B, the *S.* Pullorum load of the *S.*P + Phg group decreased significantly (*p* < 0.05). Meanwhile, the isolation rate of *S.* Pullorum in the *S.*P group significantly decreased from 55% (6/11) to 11% (1/10) (*p* < 0.05) (Table 1). In addition, the *S.* Pullorum load of liquid whole eggs in the *S.*P + Phg group dropped below the detection line at day 3 and remained negative for *S.* Pullorum infection during the next 4 days. However, *S.* Pullorum persisted in liquid whole eggs of the *S.*P group. We then detected the *S.* Pullorum load on the eggshells; Figure 6C shows that there was a relative 93-fold decrease in the *S.* Pullorum load in the *S.*P + Phg group compared to that of the Con group (*p* < 0.01), and the isolation rate of *S.* Pullorum decreased significantly from 82% (9/11) in the Con group to 30% (3/10) (*p* < 0.01) (Table 1).

To examine the effect of phage treatment on poultry breeding environments, the total number of bacteria in the air were detected at day 3 and day 6. As shown in Figure 6D, *Salmonella* infection caused an approximately 32.5-fold and 23.3-fold increase in the total bacteria load in the air at both day 3 and day 6, respectively, while phage treatment significantly decreased the total number of bacteria at 3 d (*p* < 0.001) and 6 d (*p* < 0.0001), respectively.

## 4. Discussion

Here, we confirmed that phage CKT1 could reduce the number of *S.* Pullorum in both the productive organs and various tissues, thus effectively controlling the risk of vertical transmission to eggs. We also demonstrated that phage treatment significantly inhibited the total number of bacteria in the air, thus further protecting the eggshell from *Salmonella* contamination. In addition, the reduction in *S.* Pullorum on the eggshell might prevent *Salmonella* penetration to eggs, resulting in a reduced risk of vertical transmission [30,31].

After the invasion of host cells, *S.* Pullorum can mediate the formation of *Salmonella*-containing vacuoles (SCVs) in macrophages for immune escape, leading to the liver, spleen, and other organs being colonized by *S.* Pullorum [32]. However, colonization of *S.* Pullorum in the spleen was lower than in the liver, ovary, and oviduct, probably due to the role of the immune system in eliminating *S.* Pullorum [24]. In addition, the sustained decrease in *S.* Pullorum in the heart of adult chickens in the *S.*P group was the same as the result found by Wigley et al. and Shen et al., which showed that *S.* Pullorum colonization in the heart would disappear with infection time [1,14]. However, our previous study showed that phage CKT1 could not significantly reduce the load of *S.* Pullorum in chick liver and spleen on day 6 post-infection [33]. Successive phage treatments are likely necessary to reduce *Salmonella* loads in organs. The absence of *S.* Pullorum in the ovary and oviduct for the use of phage CKT1 indicated that phage therapy was able to control the colonization level of *S.* Pullorum in the reproductive system. Currently, little is known about how phages cross the intestinal barrier and reach other organs including the reproductive tract of chickens. Despite a recent study showing that the oral administration of phage had a minimal effective on systemic dissemination in mice [12], but for the models infected with the phage-host, it appears that *S.* Pullorum in the cecum might provide a carrier for phage CKT1 to be transported across the intestinal barrier to enter macrophages by invading *Salmonella* cells, thus being successfully transferred to other organs [34]. In addition, uptake and internalization of phages may happen in a variety of epithelial cells such as those in the cervix, lung, and colon [35]. Thus, the transcytosis of phages is a natural and ubiquitous process that provides a mechanistic explanation for the occurrence of phages in other tissues [36]. A possible explanation for the elimination of *S.* Pullorum in the reproductive organs was that phage CKT1 reached the reproductive tract relying on transcytosis or on the bacterial host acting as a vector. The recently discovered phage SHWT1 [37] appears to have the ability of elimination against the intracellular colonization and formed biofilm of *Salmonella*, implying that phages can be used to clear the *S.* Pullorum colonized in macrophage. So, another possible explanation is that phage treatment reduced the load of *S.* Pullorum in the macrophage of liver and spleen, which might lead to a failed transfer of *Salmonella* to the productive organs.

At present, phage therapy has provided a relatively new tool for addressing diseases caused by bacterial infections in humans, animals, and plants [38]. However, most phages are highly host-specific, and the suitability of phage therapy for acute infections should be evaluated based on host immune system, etc., compared to antibiotics. In this study, phage CKT1 is highly specific for several *Salmonella* serovars and has a narrower host range than *Salmonella* phage STP4-a and PVP-SE1 [39,40]. Currently, numerous studies show that phages not only indirectly affect the health of the host by killing bacteria but also combat pathogens by activating the direct effect of the host’s immune system [41,42]. It has been pointed out that neutrophil-phage synergy is essential for the resolution of pneumonia [43]. Majewska et al. showed that phage treatment by oral administration induced specific antibodies in the blood of mice (IgM, IgG, IgA) [44]. In this study, we observed that phage treatment reduced the serum *Salmonella*-specific IgG antibody levels. However, antibody can usually exist in the blood for 2–12 weeks; this confusing result and the reason for reduced *Salmonella*-specific IgG antibody level should be explored further.

In addition, we found that *Salmonella* infection caused an increase in the total number of bacteria in poultry breeding environments, and phage therapy significantly reduced the total bacterial load. We previously demonstrated that *S.* Pullorum infection could result in remarkably increased *Escherichia-Shigella* and *Klebsiella* becoming the predominant bacterial taxa, while phage CKT1 treatment significantly reduced their populations in intestine and promoted the proliferation of beneficial microbiota in *Firmicutes*, thereby further alleviating the body weight loss of infected chicks [36]. So, we suppose that an abnormal intestinal microbiome may cause bacteria in feces to contain many *Salmonella* and *Escherichia* that easily form aerosol particles, and phage lysis on *Salmonella* in cecal contents and restoration of normal gut flora reduced the number of these bacteria in feces, which in turn leads to a decrease in total bacteria load in the poultry breeding environment. Another thing worth noting is that the chicken breeders without any *Salmonella* contamination used in this study were so expensive that only five adult chickens at the onset of sexual maturity in each group, which might not be sufficient to provide convincing analytical results. The present study provides a glimpse into the possibility of phage treatment on the vertical transmission of *Salmonella*, and related research still need further, in-depth investigation.

## 5. Conclusions

Altogether, our data demonstrated that phage therapy could effectively reduce the colonization level of *S.* Pullorum in both the productive organs and various tissues of adult broiler breeders, thus decreasing the probability of vertical transmission of *S.* Pullorum.

## Figures and Tables

**Figure 1 biology-12-00312-f001:**
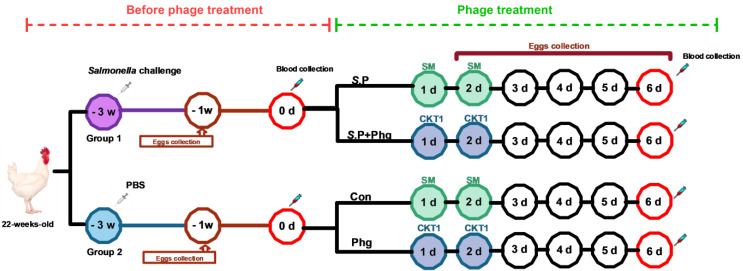
Schematic depicting the experimental design and timeline. For the groups at −3 week, 10 chickens were used in each group; For the groups at 1 d, 5 chickens were used in each group. The eggs were collected at −1 week and 2–6 d; serums were collected at 0 d and 6 d; the tissues were collected at 6 d; the total bacterial load in the air was collected at 3 d and 6 d. Group 1 or *S.*P, only *Salmonella* challenged; Group 2 or Con, control group; Phg, only phage treated; *S.*P + Phg, *Salmonella* challenged and phage treated.

**Figure 2 biology-12-00312-f002:**
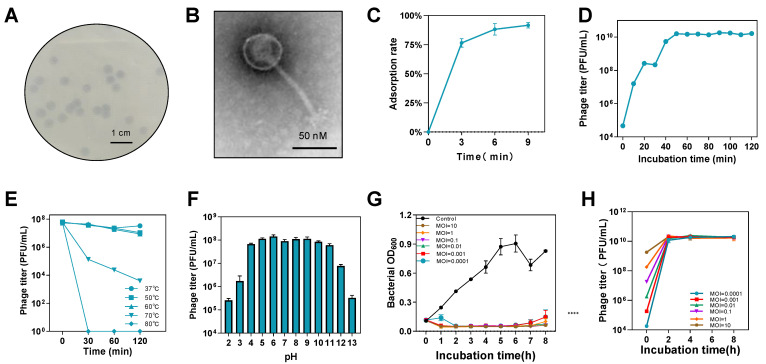
Characteristics of phage CKT1 and bacteriostatic ability in vitro. (**A**) Plaque morphology of phage CKT1 against *S.* Pullorum CVCC526. (**B**) Morphology of phage CKT1. (**C**) Adsorption rate of phage CKT1. (**D**) One-step growth curve of phage CKT1. (**E**) Temperature sensitivity of phage CKT1. (**F**) pH sensitivity of phage CKT1. (**G**) The bacteriostatic ability of phage CKT1 at different MOIs. (**H**) Phage CKT1 titer at different MOIs. Means of results from three independent assays are shown, and error bars represent the standard deviations.

**Figure 3 biology-12-00312-f003:**
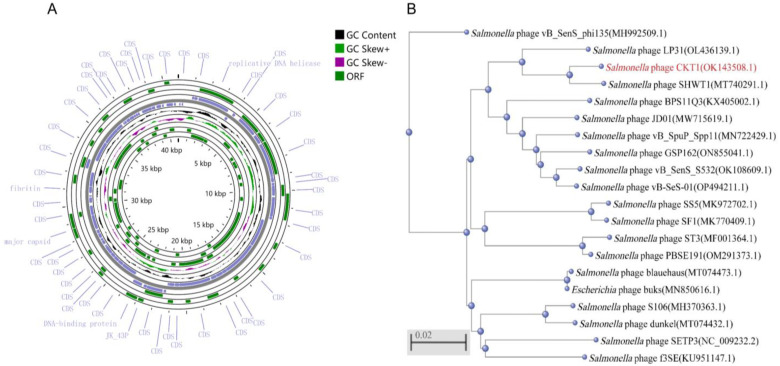
Genome analysis of phage CKT1. (**A**) Genome map of phage CKT1. (**B**) Phylogenetic tree based on the genome sequence of phage CKT1 by the neighbor-joining method.

**Figure 4 biology-12-00312-f004:**
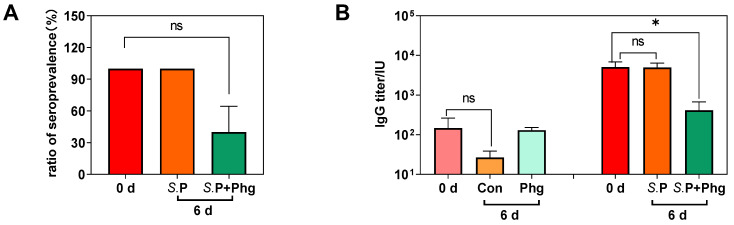
Phage CKT1 reduced the *Salmonella*-specific antibody IgG level. (**A**) The ratio of positive *Salmonella* antibodies in serum. 0 d, the serums of Group 1 at 0 d; *S.*P, the serum of *S.*P at 6 d; *S.*P + Phg, the serums of *S.*P + Phg at 6 d. (**B**) *Salmonella*-specific IgG level in serum. Con, the control group; Phg, the only phage treated group. Means of results from five independent assays are shown, and error bars represent the standard deviations. ns, *p* > 0.05 (Mann–Whitney test); * *p* < 0.05.

**Figure 5 biology-12-00312-f005:**
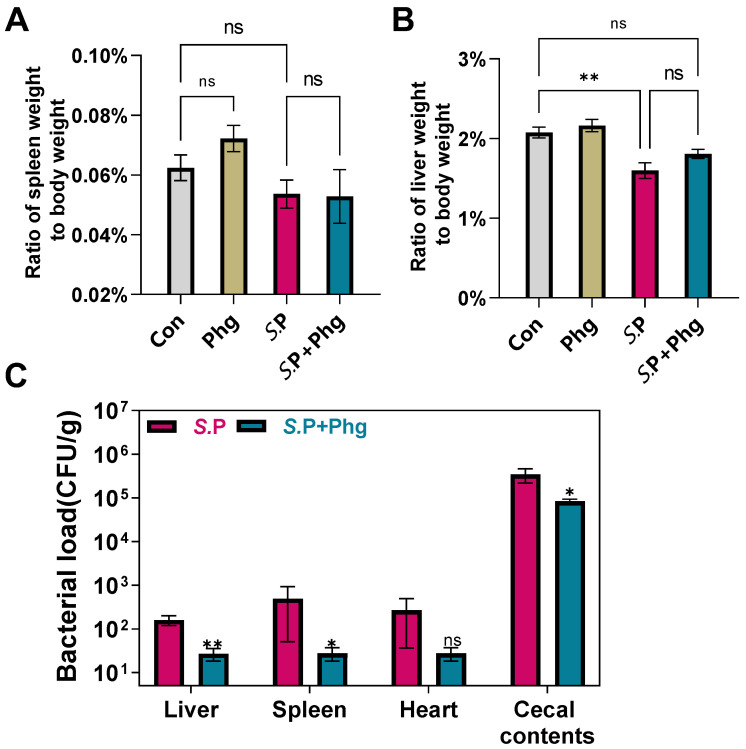
Phage CKT1 reduced the *S.* Pullorum load in various tissues of adult breeders. The ratio of spleen weight to body weight (**A**) and the ratio of liver weight to body weight at day 6 (**B**). ns, *p* > 0.05 (ordinary one-way ANOVA and post hoc Tukey’s test); ** *p* < 0.01. (**C**) *S.* Pullorum load in tissues at day 6. *n* = 5. ns, *p* > 0.05 (Mann–Whitney test); * *p <* 0.05; ** *p* < 0.01. *S*.P, only *Salmonella* challenged; Con, the control group; Phg, only phage treated; *S*.P + Phg, *Salmonella* challenged and phage treated.

**Figure 6 biology-12-00312-f006:**
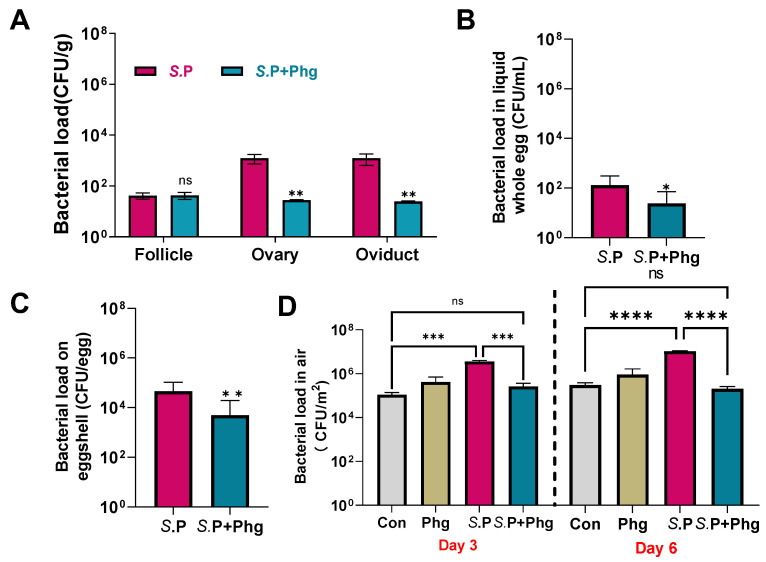
Phage CKT1 controlled the vertical transmission of *S.* Pullorum. (**A**) *S.* Pullorum load in tissues at 6 d. *n* = 5. (**B**) *S.* Pullorum load in liquid whole eggs at 2–6 d. *n* = 11 at *S*.P and *n* = 10 at *S.*P + Phg. (**C**) *S.* Pullorum load on eggshell at 2–6 d. *n* = 11 in *S.*P group and *n* = 10 in *S.*P + Phg group. * *p <* 0.05 (Mann–Whitney test); ** *p* < 0.01; ns, no significance. (**D**) Bacterial load in the air at day 3 and day 6. *n* = 3. Error bars represent the standard deviations. *** *p <* 0.001 (One-way ANOVA and post hoc Tukey’s test); **** *p <* 0.0001; ns, no significance. *S.*P, only *Salmonella* challenged; Con, the control group; Phg, only phage treated; *S.*P + Phg, *Salmonella* challenged and phage treated.

**Table 1 biology-12-00312-t001:** Number of S. Pullorum-positive samples on the total of collected laid eggs.

Sample Source from Groups	Group 1	*S.*P Group	*S.*P + Phg Group
Eggshell	6/9	9/11	3/10
Liquid whole egg	4/9	6/11	1/10

Group 1, *Salmonella* challenged before phage treatment; *S.*P group, *Salmonella* challenged during phage treatment; *S.*P + Phg group, *Salmonella* challenged and phage treated. The eggs in Group 1 were collected at −1 week and the eggs of both *S.*P group and *S.*P + Phg group were collected at 2–6 d.

## Data Availability

The complete genome sequences of phage CKT1 were deposited in the NCBI database under the GenBank accession number OK143508.

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
