# Peer review of "Salmonella* Phage CKT1 Effectively Controls the Vertical Transmission of *Salmonella* Pullorum in Adult Broiler Breeders"

_biology, 2023, doi:10.3390/biology12020312_

Round 1

Reviewer 1 Report

The manuscript entitled “Salmonella phage CKT1 effectively controls the vertical transmission of Salmonella Pullorum in adult broiler breeders" is a very interesting and well-organized research article. In general, the work is devoted to an important problem and the authors could successfully isolate and characterize Salmonella phage CKT1 with lysis ability against several S. enterica serovars. They also showed that phage CKT1 can play an active role in the control of systemic chronic Salmonella infection and eliminate its colonization in adult chickens, as demonstrated by reduced Salmonella-specific IgG level in the serum of infected chickens. Figures and Tables are clear and well presented. References are adequate and updated. I consider this manuscript of interest but after minor revision:

1.      The authors must spot alight in the discussion sections on the limitations of Phage therapy including the limited spectrum of the phage, its suitability for acute infections compared to antibiotics, and dose adjustment.

2.      Line 202: modify to “an average adsorption rate of 76% …..”

Author Response

请参阅附件。

Reviewer 2 Report

The topic of this draft is interesting but the draft manuscript has many criticisms. In primis, the phage the authors isolated are not deeply characterized for checking the absence of lysogeny. In phage therapy only strictly lytic phages are desired, and this proof is not addressed in the paper.

Second, the in vivo experimental test is very badly described. No ethical statement is provided for animal wealfare treatment. It is not clear if the authors have asked for any authorization for the use of animals. Moreover, the number of only n.5 animals for experimental unit cannot be sufficient for giving any statistically significant results.

More comments are  reported in the revised document attached

Author Response

请参阅附件。

Reviewer 3 Report

Attached

Author Response

请参阅附件。

Round 2

Reviewer 2 Report

The draft was extensively edited and revised. More major amendaments are required. Please, address all the comments in the revised doc.

Author Response

Response to the reviewer

We thank this reviewer for the time spent in reviewing and his rigorous thoughts on improving the manuscript to be better. In addition, we have carefully checked the entire article and made some revisions to unclear points in the article. Our detailed responses were listed in the following paragraphs, and other revisions were marked up using the “Track Changes” function.

Line 93 Number of sample repetitions are not reported. Did the author perform repetitions in the analysis? In which of them? please, report somewhere here or in statistic paragraph 2.11.

Sorry for the missing. We added “All above experiments were performed with three independent repetitions.” in 2.4. Phage characterization assay of Method on line 151. In addition, related information was also included in the legends of all related figures.

Line 137 The precipitated cells were resuspended with LB medium, Volume?

Thanks for the suggestion. More detailed description was added on line 135-139, and this sentence was revised to “The precipitated cells were resuspended with the same volume of fresh LB medium.”

Line 168 It is strongly recommended to report the scheme of the experimental design (Fig. 3A) in this paragraph, in order to help readers to follow. Place the scheme as Fig. 1 and rename all the Figs accordingly 

Thanks for the great suggestion. The scheme of the experimental design had been included as Figure 1 in 2.6 of the method. And all figures were renamed accordingly.

Line 181 (time -3w)

Thanks for the suggestion. Added as suggested.

Line 190 the times designated as day 1 and day 2 (Fig. 3A)

Thanks for the suggestion. Changed accordingly.

Line 194 “bacterial free” changed to “sterile”

Thanks for the suggestion. Changed accordingly.

Line 201 add “(Fig. 3A)”

Thanks for the suggestion. Added as suggested.

Line 209 adding reference

Thanks for the suggestion. Added as suggested

Line 243 volume in ml?

Thanks for pointing it out. Provided as suggested.

Line 252 Results of statistics is missing.

Results of statistics were included in the legends of all related figures.

Line 257 Table A1?

We are sorry for the mistake, and thanks for catching the error. We corrected accordingly.

Line 267 30min?

Thanks for pointing it out. To make it more clearly, this sentence was revised to “The thermal sensitivity tests showed that phage CKT1 was stable at 4~60°C but decreased at 70°C in 120 min. When the temperature increased to 80°C, phage CKT1 was almost completely inactivated in 30 min (Figure 2E).”

Lin 269 pH 12-13

Thanks for the suggestion. Changed accordingly.

Line 273 can should be could

Thanks for the suggestion. Changed accordingly.

Line 273 added “replication activity”

Thanks for the suggestion. We changed to “indicating its strong replication ability and lytic activity against S. Pullorum…”.

Line 268 missing, Maybe 2A? 2B?

We are sorry for the mistake. Corrected!

Line 294 to 298 To be placed in MeM

Thanks for the suggestion. We placed these sentences to MeM with a little modification accordingly.

Figure 3A “before the use of phage” changed to “before phage treatment”, “after the use of phage” changed to “phage treatment”

Thanks for the great suggestion. Changed accordingly.

Table 2 the title changed to “Number of S. Pullorum positive samples on the total of collected laid eggs”

Sorry for the confusion. Changed accordingly.

Line 359 Title to be edited. Results should not be reported in title.

Thanks for the suggestion. We revised the tile to “Effects of phage CKT1 on bacterial load in reproductive system, eggs, and breeding environment”

Line 394 To my previous comment: "....moreover, the number of only n.5 animals for experimental unit cannot be sufficient for giving any statistically significant results....",

the authors replied: "......We also revised the statistical analysis and this issue was added to the Discussion". Unfortunately I did not see this issue related to the weak number of animals used in the experimental units anywhere. Please, address this comment.

Sorry for the missing. We added in the last paragraph with “Another thing worth noting is that the chicken breeders without any Salmonella contamination used in this study were so expensive that only 5 adult chickens at the onset of sexual maturity in each group might not be sufficient to provide convincing analytical results. The present study provides a glimpse into the possibility of phage treatment on the vertical transmission of Salmonella, related research still need further, in-depth investigation.”.

Line 429 microphage should be macrophage?

Sorry for the mistake. Changed.

Line 431 added “relatively”, and “technology” should be changed to “tool “

Thanks for the suggestion. Changed accordingly.

Round 3

Reviewer 2 Report

Paper extensively improved.

Only 3 editings to be applied:

Line 281: the range for thermal sensitivity should be 37-60°C, not 4-60°C, as it appears to have been done in MeM

Line 311 and line 337: Paragraphs' title. See the notes in the text. The notes were already placed in R1 and the editings were previously suggested by this reviewer

Author Response

Paper extensively improved. Only 3 editings to be applied.

We would like to take this opportunity to express our appreciation to this reviewer for his great suggestions and to express our respect for his conscientious and responsible attitude. Our detailed responses were listed below, and other revisions were marked up using the “Track Changes” function.

Line 281: the range for thermal sensitivity should be 37-60°C, not 4-60°C, as it appears to have been done in MeM.

Sorry for the mistake. Changed accordingly.

Line 311 and line 337: Paragraphs' title. See the notes in the text. The notes were already placed in R1 and the editings were previously suggested by this reviewer.

Sorry for not editing them in the revised manuscript. We revised the title “3.3. Effects of phage treatment on the Salmonella-specific antibody in chickens” and “3.4. Effects of phage treatment on the bacterial load in various tissues”.